# A framework for deriving analytic steady states of biochemical reaction networks

Bryan S. Hernandez [1,2], Patrick Vincent N. Lubenia [3], Matthew D. Johnston[4], Jae Kyoung Kim [1,5]*

1 Biomedical Mathematics Group, Pioneer Research Center for Mathematical and Computational Sciences, Institute for Basic Science, Daejeon, Republic of Korea, 2 Institute of Mathematics, University of the Philippines Diliman, Quezon City, Philippines, 3 Systems and Computational Biology Research Unit, Center for Natural Sciences and Environmental Research, Manila, Philippines, 4 Department of Mathematics and Computer Science, Lawrence Technological University, Southfield, Michigan, United States of America, 5 Department of Mathematical Sciences, KAIST, Daejeon, Republic of Korea

* jaekkim@kaist.ac.kr

**Data Availability Statement:** All relevant data are within the manuscript and its Supporting information files. The computational package is available at https://github.com/Mathbiomed/COMPILES.

## Abstract

The long-term behaviors of biochemical systems are often described by their steady states. Deriving these states directly for complex networks arising from real-world applications, however, is often challenging. Recent work has consequently focused on network-based approaches. Specifically, biochemical reaction networks are transformed into weakly reversible and deficiency zero generalized networks, which allows the derivation of their analytic steady states. Identifying this transformation, however, can be challenging for large and complex networks. In this paper, we address this difficulty by breaking the complex network into smaller independent subnetworks and then transforming the subnetworks to derive the analytic steady states of each subnetwork. We show that stitching these solutions together leads to the analytic steady states of the original network. To facilitate this process, we develop a user-friendly and publicly available package, COMPILES (COMPutIng ana-Lytic stEady States). With COMPILES, we can easily test the presence of bistability of a CRISPRi toggle switch model, which was previously investigated via tremendous number of numerical simulations and within a limited range of parameters. Furthermore, COMPILES can be used to identify absolute concentration robustness (ACR), the property of a system that maintains the concentration of particular species at a steady state regardless of any initial concentrations. Specifically, our approach completely identifies all the species with and without ACR in a complex insulin model. Our method provides an effective approach to analyzing and understanding complex biochemical systems.

## Author summary

Steady states often describe the long-term behaviors of biochemical systems, which are typically based on ordinary differential equations. To derive a steady state analytically, significant attention has been given in recent years to network-based approaches. While this approach allows a steady state to be derived as long as a network has a special structure,

**Funding:** BSH and JKK are supported by the Institute for Basic Science IBS-R029-C3. MDJ is supported by NSF grant DMS-2213390. The funders had no role in study design, data collection and analysis, decision to publish, or preparation of the manuscript.

**Competing interests:** The authors have declared that no competing interests exist.

complex and large networks rarely have this structural property. We address this difficulty by breaking the network into smaller and more manageable independent subnetworks, and then use the network-based approach to derive the analytic steady state of each subnetwork. Stitching these solutions together allows us to derive the analytic steady state of the original network. To facilitate this process, we develop a user-friendly and publicly available package, COMPILES. COMPILES identifies critical biochemical properties such as the presence of bistability in a genetic toggle switch model and absolute concentration robustness in a complex insulin signaling pathway model.

## Introduction

Chemical reaction networks (CRNs) are fundamental in disciplines such as systems biology [1–4] and industrial chemistry [5]. The dynamical behavior of such networks is frequently modeled using a system of ordinary differential equations (ODEs) based on mass-action kinetics [6]. It is particularly important to determine properties of the steady states of ODEs because they often describe long-term behaviors of the CRNs. In practice, however, this is made challenging by the high-dimensionality, nonlinearities, and typically unknown parameter values of such systems. These factors make traditional tools such as numerical studies and bifurcation analysis impractical or impossible.

Significant attention has consequently been given in recent years to CRNs with special structures in their underlying interaction networks, such as being weakly reversible (WR) and deficiency zero (DZ) [7]. A CRN is WR if the network is the union of reaction cycles, while the deficiency is a nonnegative integer which measures the dependency of the reactions. It is known that, regardless of the dimension of the system or the rate parameter values, the mass-action system associated with a WR and DZ network admits a unique locally stable steady state within each positive stoichiometric class [8–10]. The steady state set is furthermore known to have a monomial parametrization, i.e., a monomial with free parameters as variables, which can be constructed directly from the reaction graph [11–13]. The construction of such parametrizations has been vital in the study of absolute concentration robustness, which is the capacity for a system to have a species whose steady state value is robust to changes in initial conditions [14–16], and multistationarity, which is the capacity of a network to have multiple stoichiometrically-compatible steady states [17, 18].

In general, however, CRNs arising in applications are rarely WR or DZ. Hence, the method of *network translation*, which can modify the network structure while maintaining the dynamics, has been developed [13, 16, 19–21]. It involves shifting reactions, i.e., swapping reactions for ones with the same stoichiometric differences while keeping the original reaction rates. For instance, a reaction $A + B \rightarrow B$ could be shifted to $A \rightarrow 0$ while we keep the rate of the original reaction $k_1 ab$ to maintain the original dynamics. This network translation leads to a *generalized chemical reaction network* (GCRN) with two associated structures. The first structure is the network's *stoichiometric CRN* with nodes that include the new stoichiometric nodes $A$ and 0 from the shifted reaction, and the *kinetic-order CRN* with nodes that include the original source node $A + B$ from the original reaction. If both stoichiometric and kinetic-order networks are WR and DZ, then the translated GCRN is WR and DZ. In this case, a parametrization of the steady states can be computed easily [20]. In practice, however, the process of network translation can be challenging for two reasons: (a) it can be difficult to find a WR and DZ network translation; (b) even after the network translation, the two desired structures are rarely satisfied, in particular for large and complex networks [22, 23].

In this paper, we address these challenges by breaking the process of network translation and computation of steady-state parametrization into smaller and more manageable pieces. We start by decomposing the CRNs into stoichiometrically independent subnetworks [24, 25]. We then translate the subnetworks into WR and DZ GCRNs by shifting reactions. If successful, we derive the steady states of each translated subnetwork independently. The steady state of the original CRN can then be derived by stitching together the steady states of the subnetworks. To facilitate this process, we have also developed a computational package called COMPILES (COMPutIng anaLytic stEady States). We demonstrate the utility of our approach on examples drawn from a CRISPRi toggle switch model [26] and a metabolic insulin signaling model [27]. In particular, we identify their critical properties such as bistability and absolute concentration robustness without an enormous number of numerical calculations, unlike previous studies.

## Results

### Derivation of analytic steady states via network decomposition and network translation

To illustrate the approach of deriving analytic steady states via network decomposition and network translation, we consider a simple CRN ($\mathcal{N}$) (Fig 1a upper left). This network is not WR because the reaction $B + C \to B$ is not contained in a cycle. Furthermore, it is not DZ because it has six nodes ($n = 6$), two connected components ($\ell = 2$), and the rank of its stoichiometric matrix is three ($s = 3$) (lower left). This gives a deficiency of $\delta = n - \ell - s = 6 - 2 - 3 = 1 \neq 0$ (see the Methods Section for details of the deficiency). We decompose this network into two *independent subnetworks* as shown in Fig 1a (upper right) (see the Methods Section for details on independent subnetworks). Specifically, we group the reactions of the network in such a way that the rank of the stoichiometric matrix of the original network (lower left) equals the sum of the ranks of the stoichiometric matrices of the resulting subnetworks (lower right).

The first subnetwork ($\mathcal{N}_1$) (Fig 1b left) is a WR and DZ network. On the other hand, the second subnetwork ($\mathcal{N}_2$) (Fig 1b middle) is neither WR nor DZ. Thus, we have to translate $\mathcal{N}_2$ to make it WR and DZ. Specifically, $\mathcal{N}_2$ is not WR because the reaction $B + C \to B$ does not belong to a cycle. Thus, we replace it with $A + C \to A$, having the same stoichiometry to obtain the network, via the process of network translation, referred to as the *stoichiometric CRN* and denoted by $\mathcal{N}_2'^S$ (Fig 1b middle) (see the Methods section and [13] for details on network translation). We keep the original reaction rate ($k_2bc$) for $A + C \to A$ so that the dynamics does not change. However, as the network structure changes, and in particular, as all the reactions now belong to a cycle, $\mathcal{N}_2'^S$ is WR. Furthermore, the deficiency of the stoichiometric CRN, called the effective deficiency, is zero, i.e., EDZ, because it has three nodes ($n = 3$), one connected component ($\ell = 1$), and a stoichiometric matrix of rank two ($s = 2$) so that $\delta = 3 - 1 - 2 = 0$.

Next, we construct the *kinetic-order CRN*, which is denoted by $\mathcal{N}_2'^K$. To do this, we first take all the edges of $\mathcal{N}_2'^S$. Then, we fill 0 in the tail of the edge associated with $k_2$ (Fig 2a (iii)) because the source node of the edge associated with $k_2$ in the original network ($\mathcal{N}_2$) is 0 (Fig 2a (i)). Similarly, we fill $A$ in the tail of the edge associated with $k_3$ (Fig 2a (iii)). Because the source nodes of the edges with $k_1$ and $k_4$ are $A + C$ and $B + C$ in the original network (Fig 2a (i)), two source nodes ($A + C$ and $B + C$) are now placed in a single node (Fig 2a (iii)). To separate the two source nodes, a phantom edge, which is an edge with a free parameter ($\sigma_1$), is introduced in such a way that the resulting network is still WR (Fig 2a (iv)) (see the Methods section and [21] for details on phantom edges). For the phantom edge, a zero stoichiometric vector is

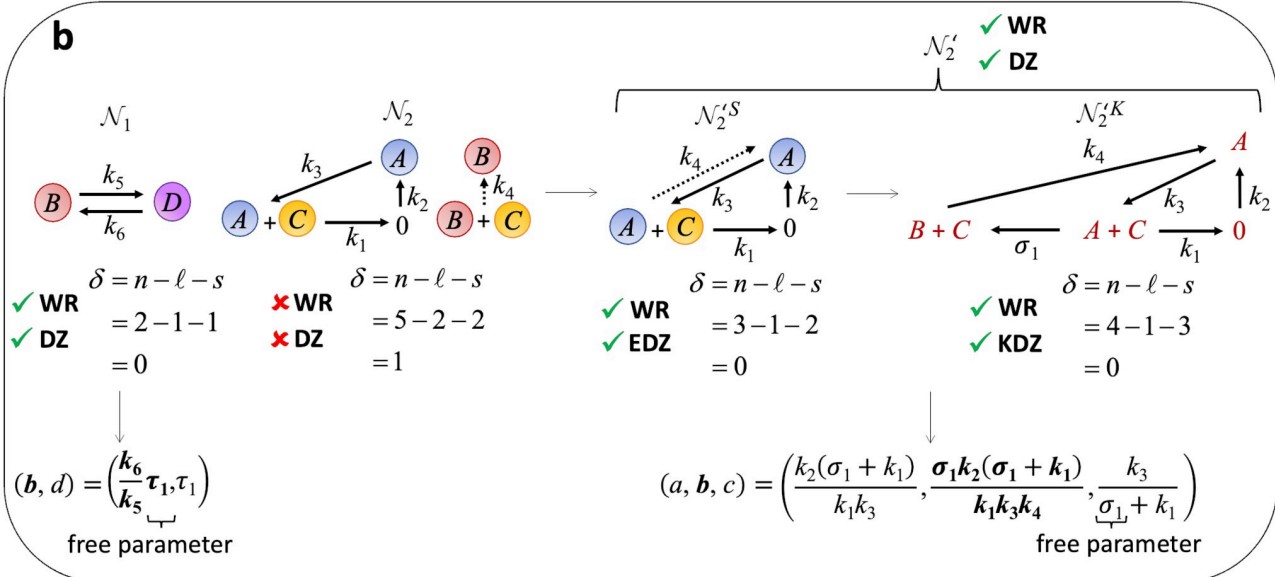

**Fig 1. Derivation of analytic steady state via network decomposition. a** The CRN ($\mathcal{N}$) is decomposed into independent subnetworks ($\mathcal{N}_1$ and $\mathcal{N}_2$) in such a way that the rank of the original matrix ($s$) is equal to the sum of the ranks of the stoichiometric matrices of subnetworks ($s_1 + s_2$). **b** Since the subnetwork $\mathcal{N}_2$ is not WR and DZ, network translation is performed. The translated subnetwork ($\mathcal{N}_2'$) is WR and DZ, i.e., its stoichiometric and kinetic-order CRNs ($\mathcal{N}_2'^S$ and $\mathcal{N}_2'^K$, respectively) are both WR and DZ, while its dynamics is equivalent to the dynamics of the original subnetwork ($\mathcal{N}_2$). Then, the steady states of the original subnetwork ($\mathcal{N}_1$) and the translated subnetwork ($\mathcal{N}_2'$) can be analytically derived since they are WR and DZ (see Fig 2 for details). **c** The steady states of subnetworks are merged to identify the steady states of the original networks. In particular, the steady states of the common species of $\mathcal{N}_1$ and $\mathcal{N}_2$ (i.e., species $B$) are equated, which eliminates the free parameter $\tau_1$. Then, by combining the steady state of every species, the steady state of the original network can be derived with one remaining free parameter $\sigma_1$. This $\sigma_1$ is computed from the conserved quantity in the network, which is the sum of the initial concentrations of species $B$ and $D$ (i.e., $b_0 + d_0$).

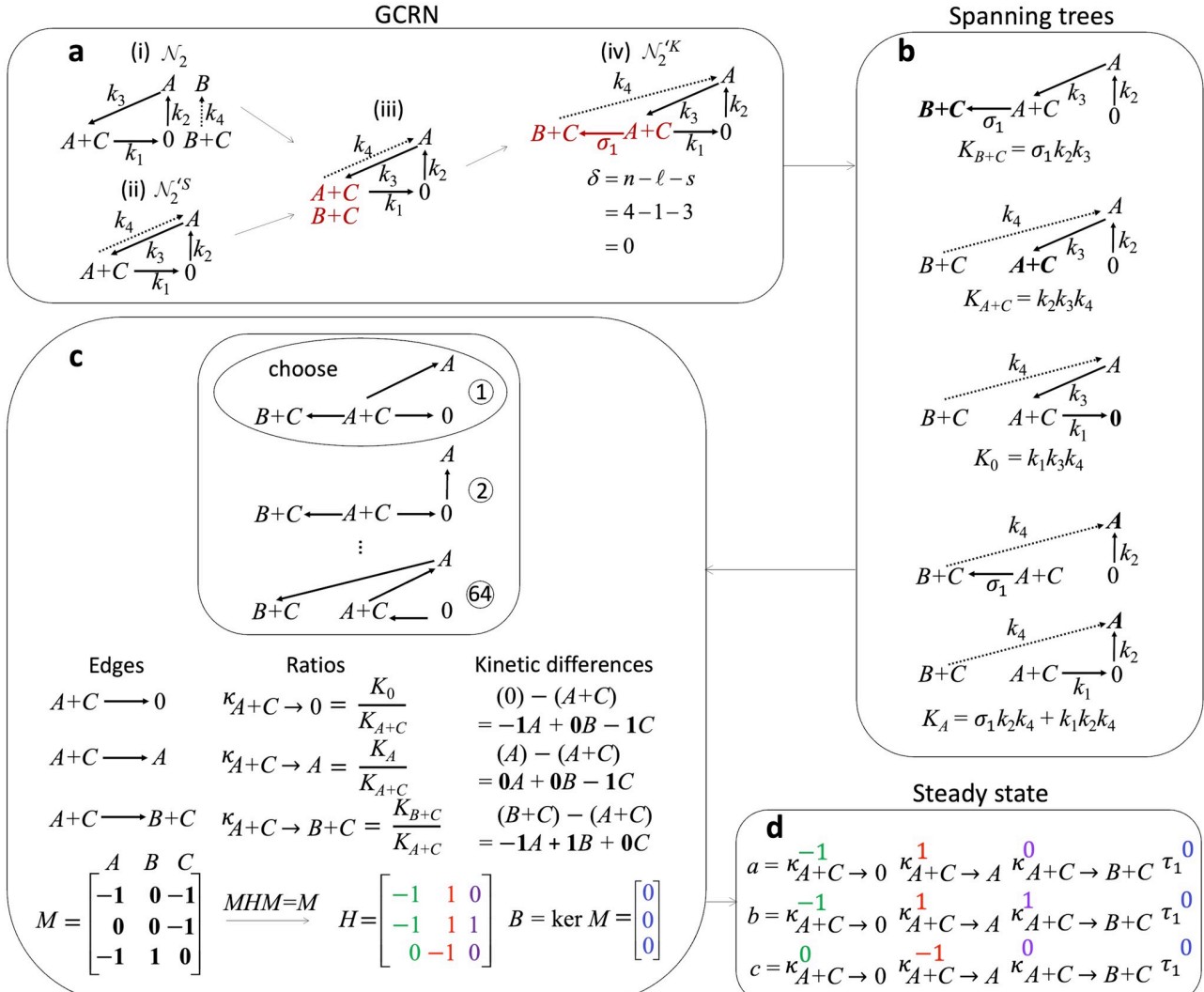

**Fig 2. Derivation of analytic steady state via network translation. a** The kinetic-order CRN $\mathcal{N}_2'^K$ is constructed from the edges of the stoichiometric CRN $\mathcal{N}_2'^S$ but the nodes are the source nodes of the associated reactions in the original network $\mathcal{N}_2$. For instance, the edges associated with $k_1$ and $k_4$ in $\mathcal{N}_2'^K$ (ii) have corresponding source nodes $A + C$ and $B + C$, respectively, in the original network (i). Since there are two source nodes ($A + C$ and $B + C$) which share a single node in (iii), a phantom edge with a zero stoichiometric vector and a free parameter rate constant ($\sigma_1$) is introduced (upper right). Notice that this operation maintains the dynamics of the original network. Proceed to the next step since the deficiency of $\mathcal{N}_2'^K$ is zero (i.e., KDZ). **b** Collect all the spanning trees of $\mathcal{N}_2'^K$ (i.e., connected subgraphs of the $\mathcal{N}_2'^K$ without a cycle) which direct towards each node ($B + C$, $A + C$, 0, and $A$). Then multiply the rate constants associated with the edges of each spanning tree. For instance, the products of the rate constants associated with the edges of the two spanning trees towards node $A$ are $\sigma_1 k_2 k_4$ and $k_1 k_2 k_4$ (bottom). Compute each tree constant ($K_i$) by adding the product of the rate constants over all the spanning trees towards node ($i$). Hence, the tree constant $K_A = \sigma_1 k_2 k_4 + k_1 k_2 k_4$ is obtained. **c** Choose an arbitrary tree (i.e., a connected graph without a cycle) containing all the nodes in $\mathcal{N}_2'^K$, which is not necessarily its subgraph. Then, for each edge ($i \to i'$) of the chosen tree, find the ratios of the tree constants $\left( \kappa_{i \to i'} = \frac{K_{i'}}{K_i} \right)$ and the kinetic difference ($i' - i$) (middle). For instance, the edge $A + C \to 0$ has the ratio of the tree constants $\kappa_{A+C \to 0} = \frac{K_0}{K_{A+C}}$ and kinetic difference $(0) - (A + C) = -1A + 0B - 1C$. From the kinetic differences, construct the matrix $M$ (bottom) by listing in each row the coefficients in the kinetic difference associated with an edge of the tree. Hence, the coefficients in the kinetic difference $-1A + 0B - 1C$ are the entries of the first row of $M$ (i.e., $[-1, 0, -1]$). Then, compute a generalized inverse $H$ of $M$ (i.e., $MHM = M$) and the kernel $B$ of $M$. **d** Derive the analytic steady state of the network from the ratio of the tree constants $\kappa_{i \to i'}$ and the matrices $H$ and $B$. Specifically, raise the ratios of the tree constants $\kappa_{i \to i'}$ in a component-wise manner to the entries of a row of $H$, and get their product. For instance, the ratio of the tree constants $\kappa_{A+C \to 0}$, $\kappa_{A+C \to A}$, and $\kappa_{A+C \to B+C}$ are raised to the entries in the first row of $H$ ($-1$, 1 and 0, respectively). This gives $\kappa_{A+C \to 0}^{-1} \kappa_{A+C \to A}^1 \kappa_{A+C \to B+C}^0$. Meanwhile, one free parameter ($\tau_1$) is introduced because the number of column of $B$ is one. This $\tau_1$ is raised to the entry of a row in $B$. Hence, $\tau_1^0$ is obtained. Finally, get the product $\kappa_{A+C \to 0}^{-1} \kappa_{A+C \to A}^1 \kappa_{A+C \to B+C}^0 \tau_1^0$, which is the steady state value of species $A$. The steady state values for species $B$ and $C$ can be computed in a similar manner.

always assigned to maintain the dynamics of the original network. This produces a kinetic-order CRN $\mathcal{N}_2'^{K}$.

Then, we compute the deficiency of $\mathcal{N}_2'^{K}$, i.e., the kinetic deficiency. The differences of the nodes of the edges except for the phantom edge are the same as the reaction vectors of the stoichiometric CRN, and the difference of the nodes of the phantom edge $((B + C) - (A + C) = [-1, 1, 0, 0]^\top)$ is independent from the reaction vectors of the stoichiometric CRN. Thus, the rank of the matrix of reaction vectors of $\mathcal{N}_2'^{K}$ is one more than the rank of the stoichiometric matrix of $\mathcal{N}_2'^{S}$ ($s = 3$). Furthermore, $\mathcal{N}_2'^{K}$ has one more node $(B + C)$ compared to $\mathcal{N}_2'^{S}$ ($n = 4$). Additionally, the number of connected components are the same ($\ell = 1$). Since $\delta = n - \ell - s = 4 - 1 - 3 = 0$, the kinetic deficiency of $\mathcal{N}_2'^{K}$ is zero (i.e., KDZ). We proceed to the next step as the kinetic deficiency is zero; otherwise, additional conditions need to be satisfied (see [20, 21]).

The stoichiometric CRN $\mathcal{N}_2'^{S}$ and kinetic-order CRN $\mathcal{N}_2'^{K}$ form the *translated network*, denoted by $\mathcal{N}_2'$, which is a generalized chemical reaction network (GCRN) (see the Methods section for details on GCRNs). $\mathcal{N}_2'$ is DZ because $\mathcal{N}_2'^{S}$ is EDZ and $\mathcal{N}_2'^{K}$ is KDZ. Furthermore, $\mathcal{N}_2'$ is WR because both $\mathcal{N}_2'^{S}$ and $\mathcal{N}_2'^{K}$ are WR. When $\mathcal{N}_2'$ is WR and DZ, the parametrization of the steady states can be easily calculated (see Theorem 2 in the Methods section).

To calculate the parametrization of the steady states, we need to get all the spanning trees of $\mathcal{N}_2'^{K}$, i.e., connected subgraphs without a cycle directed towards each node as shown in Fig 2b. For instance, there is only one spanning tree directed towards $B + C$ because there is only one connected subgraph of the kinetic-order CRN without a cycle that points towards $B + C$ (Fig 2a (iv)). Then, to get the tree constant ($K_i$) associated with a particular node $i$ with only one spanning tree, we multiply the rate constants corresponding to the edges of this spanning tree (see the description of tree constants in Theorem 2 of the Methods section and [13] for more details). Hence, we obtain $K_{B+C} = \sigma_1 k_2 k_4$ as the tree constant directed towards the node $B + C$. In general, if there is more than one spanning tree directed towards a particular node, we have to repeat the process of getting the corresponding product for each spanning tree and then get the sum of the products over all the spanning trees directed towards the particular node. Specifically, there are two spanning trees directed towards node $A$ because there are two connected subgraphs of $\mathcal{N}_2'^{K}$ without a cycle that point towards node $A$. Then, multiply the rate constants associated with the edges of each spanning tree. Hence, we obtain $\sigma_1 k_2 k_4$ and $k_1 k_2 k_4$ (bottom) as the products of the rate constants associated with the two spanning trees directed towards node $A$. Thus, the tree constant associated with node $A$ is $K_A = \sigma_1 k_2 k_4 + k_1 k_2 k_4$. Note that the term "tree constant" was introduced in [13] to refer to the formulas following from the matrix tree theorem presented in [28] and adapted to chemical reaction network theory in [11].

Next, from $\mathcal{N}_2'^{K}$ (Fig 2a (iv)), we form a tree (i.e., a connected graph without a cycle) that contains all the nodes of this CRN. The possible trees that can be formed are given in Fig 2c (top). Then, for each edge of the tree, find the ratio of the tree constants (center) and the kinetic difference (middle right), for example, the ratio of the tree constants $\kappa_{A+C \to 0} = \dfrac{K_0}{K_{A+C}}$, and the associated kinetic difference of $(0) - (A + C) = -1A + 0B - 1C$. After doing this for each edge of the tree, we construct the matrix $M$ where each row corresponds to the vector of coefficients in the kinetic difference associated with each edge. For instance, the first row of $M$ is the first kinetic difference $-1A + 0B - 1C = [-1, 0, -1]$ associated with the first edge $A + C \to 0$. In addition, the second and third rows of $M$ are precisely the vectors of coefficients in the kinetic differences associated with the two remaining edges of the tree. We then compute a generalized inverse $H$ of $M$ (i.e., $MHM = M$) and the kernel $B$ of $M$.

From the matrices $H$ and $B$ together with the ratios of the tree constants, we can derive the analytic steady state of the subnetwork $\mathcal{N}_2$ (Fig 2d). In particular, to get the analytic steady state of the first species $A$, we raise each ratio of the tree constants associated with the three edges (i.e., $\kappa_{A+C\to 0}$, $\kappa_{A+C\to A}$, and $\kappa_{A+C\to B+C}$) to each of the entries in the first row of $H$ (i.e., $-1$, 1, and 0, respectively), and then multiply them to obtain $\kappa_{A+C\to 0}^{-1} \cdot \kappa_{A+C\to A}^{1} \cdot \kappa_{A+C\to B+C}^{0}$. Additionally, we introduce a number of free parameters, as many as the number of columns of $B$. We raise this to the value of the first row of $B$ (i.e., 0) and we get $\tau_1^0$. We multiply this from the previously obtained product that gives
$(\kappa_{A+C\to 0}^{-1} \cdot \kappa_{A+C\to A}^{1} \cdot \kappa_{A+C\to B+C}^{0}) \cdot (\tau_1^0) = \kappa_{A+C\to 0}^{-1} \cdot \kappa_{A+C\to A}^{1} \cdot \kappa_{A+C\to B+C}^{0}$, which is the value of the analytic steady state of the first species $A$ with no free parameter $\tau_1$ because $\tau_1$ is raised to zero. By following the same procedure, we can get the analytic steady state values of the two remaining species of the translated subnetwork ($\mathcal{N}_2'$).

Because $\mathcal{N}_1$ is a WR and DZ network, we did not need the network translation. In this case, we can get its steady state by applying the procedure (Fig 2) to $\mathcal{N}_1$. While we illustrate the parametrization of steady states with the simple example, a rigorous description is provided in the Methods section (see Theorem 2).

Then, we combine the steady states of the two subnetworks ($\mathcal{N}_1$ and $\mathcal{N}_2$) to derive the steady state of the original whole network $\mathcal{N}$. Specifically, we equate the steady state values of the common species of the two subnetworks. That is, since species $B$ appears in both subnetworks, we equate the steady state values of species $B$ in $\mathcal{N}_1 \left( \text{i.e., } \dfrac{k_6}{k_5}\tau_1 \right)$ and $\mathcal{N}_2 \left( \text{i.e., } \dfrac{\sigma_1 k_2 (\sigma_1 + k_1)}{k_1 k_3 k_4} \right)$ (Fig 1b bottom). As a result, we get $\tau_1 = \dfrac{\sigma_1 k_2 k_5 (\sigma_1 + k_1)}{k_1 k_3 k_4 k_6}$. By using this, we can eliminate the free parameter ($\tau_1$) in the combined steady state (Fig 1c upper left) and derive the analytic steady state of the whole network $\mathcal{N}$ (Fig 1c) with one free parameter $\sigma_1$. This free parameter can be solved using the conservation law $b + d = b_0 + d_0$ where $b_0$ and $d_0$ are the initial concentrations of species $B$ and $D$, respectively. That is, by substituting the steady state values of species $B$ and $D$ to the conservation law, we can solve for the value of $\sigma_1$ in terms of the conserved quantity ($b_0 + d_0$). The closed form can be used to easily identify the critical features of a steady state, such as multistability and concentration robustness, which we will illustrate as follows.

## The analytic steady state of a simple CRISPRi toggle switch model

We now apply our method to a *CRISPRi toggle switch model* [26] with nine species (Table A in S1 Text) and 14 reactions (Fig 3a left). In the model, the deactivated mutant protein dCas9 with a single guide RNA 1 (*CS* complex) and dCas9 with a single guide RNA 2 (*CT* complex) bind to a single specific site on their target genes $H$ and $G$, respectively, called *specific binding*, which forms complexes $CS : H$ (species $P$) and $CT : G$ (species $R$). With the model, it was shown that experimentally observed bistability of species $S$ (i.e., single guide RNA 1) in response to parameter change was impossible. This leads to the identification of previously unidentified reactions of *unspecific binding* (i.e., binding to unspecific sites not matching the single guide RNA sequence, e.g., $CS$ and $CT$ complexes bind unspecifically to genes $G$ and $H$ forming complexes $CS : G$ and $CT : H$ as opposed to the formation of complexes $CS : H$ and $CT : G$ in the specific binding), which could explain the bistability in the system [26].

In a previous study, numerical simulations of the model for some parameters were used to show the absence of the bistability. In particular, in Fig 7 of the Supplementary Information of [26], the authors considered a range of values for $k_{10}$ (i.e., unbinding rate constant of the specifically bound $CT$ (dCas9 with a single guide RNA 2) to $G$ (Gene 1)) and performed numerical simulations to identify the effect of varying $k_{10}$ on the steady state concentration of $S$ (single

**Fig 3. Derivation of analytic steady state of a CRISPRi toggle switch model via network decomposition and network translation. a** A CRISPRi toggle switch network (left) that assumes dCas9 with a single guide RNA 1 ($CS$ complex) and dCas9 with a single guide RNA 2 ($CT$ complex) bind to a single site on their target genes $H$ and $G$, respectively. The CRN has nine species (i.e., $CS$, $CT$, $C$, $G$, $H$, $P$, $R$, $S$, and $T$) and 14 reactions (i.e., $R_1$, ..., $R_{14}$), which is decomposed into seven independent subnetworks ($\mathcal{N}_1, \ldots, \mathcal{N}_7$) (right). **b** As the subnetworks $\mathcal{N}_1$ and $\mathcal{N}_2$ are not WR and DZ, network translation is performed. The translated subnetworks ($\mathcal{N}_1'$ and $\mathcal{N}_2'$) are WR and DZ (i.e., both its stoichiometric and kinetic-order CRNs are

WR, and it is an EDZ and a KDZ generalized network) while their dynamics are equivalent to the dynamics of the original subnetworks ($\mathcal{N}_1$ and $\mathcal{N}_2$, respectively). Then, the steady states of the translated subnetworks ($\mathcal{N}'_1$ and $\mathcal{N}'_2$) and the original subnetworks ($\mathcal{N}_3, \ldots, \mathcal{N}_7$) can be analytically derived (see Fig 2 for the outline of the steps). **c** The steady states of the subnetworks are combined by equating the steady states of the common species. For instance, the species *CS* is common to both subnetworks $\mathcal{N}_3$ and $\mathcal{N}_5$ so the steady state values of *CS* of both subnetworks are equated to each other (left). Then after solving the steady state of the whole network, two free parameters ($\tau_1$ and $\tau_2$) are left, which can be solved in terms of the conserved quantities. This analytic steady state solution could be used to determine the behavior of the steady state concentrations of species with respect to varying rate constants. In particular, the monotonicity of the steady state concentration of species *S* over varying rate constant $k_i$, when the rest of the rate constants are set to one, is illustrated on the right.

guide RNA 1). This is time-consuming. Importantly, it was shown only for a limited choice of parameters. To circumvent this, we derive the analytic steady states of the model to show the absence of the bistability for any choice of parameters.

To do this, we decompose the model into seven independent subnetworks (Fig 3a right). Then, for subnetworks that are not WR and DZ (i.e., $\mathcal{N}_1$ and $\mathcal{N}_2$), the network translation was performed to obtain WR and DZ generalized networks. The translated subnetworks $\mathcal{N}'_1$ and $\mathcal{N}'_2$ are WR and DZ because their stochiometric networks ($\mathcal{N}'^{S}_1$ and $\mathcal{N}'^{S}_2$, respectively) are WR and EDZ and kinetic-order networks ($\mathcal{N}'^{K}_1$ and $\mathcal{N}'^{K}_2$, respectively) are WR and KDZ, while their dynamics are equivalent to the original subnetworks $\mathcal{N}_1$ and $\mathcal{N}_2$, respectively (Fig 3b left). This allowed us to derive the analytic steady state of each subnetwork (Fig 3b) using the method outlined in Fig 2. By combining the steady states of the subnetworks, we computed the analytic form of the steady state (Fig 3c upper left) with the values of the free parameters $\tau_1$ and $\tau_2$ using the conservation laws $h + p = h_0 + p_0$ and $g + r = g_0 + r_0$ where $h_0, p_0, g_0, r_0$ are the initial concentrations of species *H*, *P*, *G*, and *R*, respectively. In particular, we derived the analytic steady state of *s* as follows:

$$s = \frac{Q + \sqrt{Q^2 + 4 \cdot \dfrac{k_1 k_3 k_7 k_{12}(g_0 + r_0)}{k_4 k_8 k_{11} k_{13}}}}{2 \cdot \dfrac{k_3 k_7 k_{12}}{k_4 k_8 k_{11}}}$$

where

$$Q = -\left(1 + \frac{k_2 k_5 k_9 k_{12}(h_0 + p_0)}{k_6 k_{10} k_{11} k_{14}}\right) + \frac{k_1 k_3 k_7 k_{12}(g_0 + r_0)}{k_4 k_8 k_{11} k_{13}}.$$

From this, we can easily show that the steady state of *S* always increases as we increase $k_{10}$ because *Q* also increases as we increase $k_{10}$. This confirms the absence of the bistability of *S* against $k_{10}$, which was shown in a previous study under limited conditions [26]. Importantly, we can also easily observe from the formula that the steady state concentration of *S* increases monotonically over $k_1$, $k_6$, and $k_{14}$, while monotonically decreasing over $k_2$, $k_5$, $k_9$, and $k_{13}$. Furthermore, the monotonicity of the steady state concentration of species *S* on the rest of the rate constants can be shown analytically (see the Supplementary Methods in S1 Text for details). Hence, the bistability of *S* is impossible in this model for any choice of parameters. This is inconsistent with experimental observations, which indicates a missing mechanism and is consistent with the previous study [26].

## Computational package, COMPILES

Applying our method (Figs 1 and 2) of getting the analytic steady states of a system via network decomposition and translation is challenging for complex CRNs. To resolve this, we developed a user-friendly, open-source, and publicly available computational package, COMPILES

(COMPutIng anaLytic stEady States), (https://github.com/Mathbiomed/COMPILES) which automatically decomposes the CRN into its finest independent decomposition, translates the subnetworks that are neither WR nor DZ to WR and DZ GCRNs, parametrizes their steady states, and combines these solutions to the subnetworks to get the analytic steady state solution to the whole system. With this package, we were able to get the solutions to systems that involve as many as 35 reactions.

We illustrate how COMPILES derives the analytic steady state with an example of a complex CRN that describes the signaling cascades activated by insulin [27, 29] (Fig 4a). To use the package, one simply inputs the reactions in the system's CRN. COMPILES automatically decomposes the network into its finest independent decomposition (i.e., independent decomposition with the maximum number of subnetworks). Specifically, the package decomposes the 35 reactions of the insulin network into 10 subnetworks (Fig 4b right). Then the steady state of each subnetwork is automatically derived using the method outlined in Figs 1b and 2. COMPILES finally combines these subnetwork solutions to derive the analytic steady state solution for the entire network. It outputs the steady state solution with the free parameters and the conservation laws in the original network (Fig 4b lower left).

Using these results, we investigated the capacity for the system to exhibit *absolute concentration robustness (ACR)*. A system is said to have ACR in a species $X$ if the system has the same steady state concentration for $X$ regardless of any initial concentrations [14, 15]. It basically describes the capacity to maintain the concentration of a particular species at steady state within a narrow range, regardless of the changes in the amounts of other network species that might vary due to environmental variables in the system [14]. Previously, detecting species with ACR could be done only under limited structural network conditions [14, 30]. Here, our approach allows us to detect the ACR for large networks in a manageable fashion.

To simplify the analysis of the network, we set the rate constants to 1 and the free parameters can be solved in terms of the conserved quantities. Then, we analyzed how the steady state of each species has ACR against the five conserved quantities in the system (Fig 4c). When a species is inside a circle corresponding to a conserved quantity, it means that the steady state concentration of this species does not change even though the associated conserved quantity is varied. Interestingly, the steady states of the eight species are ACR from all five conserved quantities. In particular, one of them is $X_{20}$ (intracellular GLUT4). This indicates that the long-term concentration of GLUT4 in the cell could be maintained because the system has ACR in the species $X_{20}$. This allows the cell to have a certain level of glucose transporters that may be translocated to the cell surface whenever the cell needs glucose for energy metabolism. On the other hand, the system does not have ACR in species $X_{21}$ (cell surface GLUT4). This makes sense since the amount of GLUT4 transported to the cell surface fluctuates as the energy needs of the cell vary.

## Discussion

In this paper, we have developed a framework and a corresponding computational package which analytically derive the steady states of a large class of chemical reaction networks. This framework utilizes network decomposition to break a CRN into smaller and more manageable pieces. This is significantly more efficient than the previous methods of solving steady states presented in [20, 21], which required performing the network translation on whole networks that are not WR and DZ, which could be very challenging if the given network is large and complicated. Using our approach, we can derive the steady state of a CRN with mass-action kinetics if each independent subnetwork is either a WR and DZ network, or can be transformed into a WR and DZ generalized network. To facilitate this, we developed a user-friendly and publicly available computational package, called COMPILES.

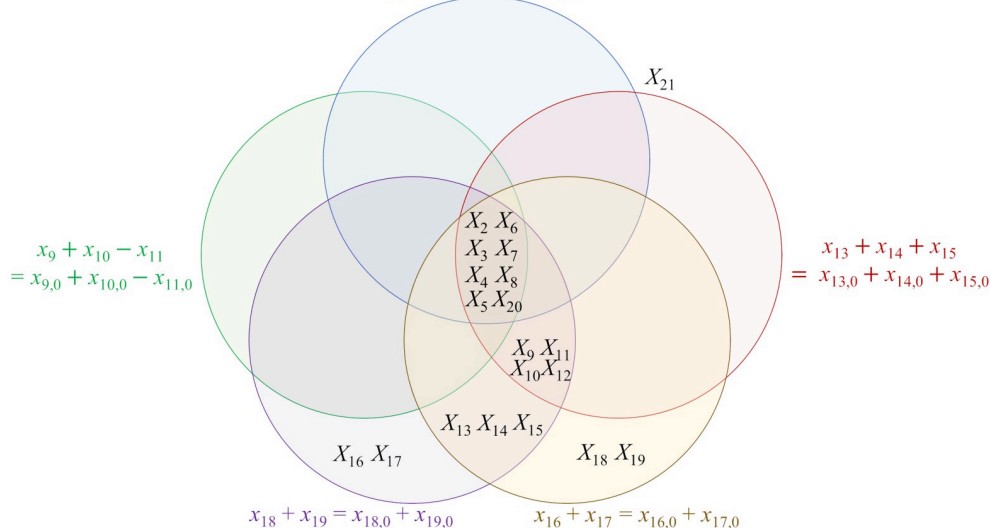

**Fig 4. Derivation of analytic steady state of an insulin signaling network via computational package COMPILES. a** The CRN of a mathematical model of insulin metabolic signaling pathway with 20 species ($X_2, \ldots, X_{21}$) and 35 reactions ($R_1, \ldots, R_{35}$). **b** A schematic diagram that describes the computational package COMPILES. As the insulin network is large, a computational package is developed to analytically derive the steady state of the network. If the users enter the reactions of the network (top left), then the package derives the steady state with the free parameters and the conservation laws of the network (bottom left). This is done by decomposing the network into independent subnetworks as outlined in Fig 1a and 1b and then deriving the steady state of each subnetwork as outlined in Fig 2 (right). Finally, the steady states of the subnetworks are combined to derive the analytic steady state of the original whole network as shown in

Fig 1c. **c** The summary of the independence of the steady state concentrations from the conserved quantities in the network when $k_1 = \cdots = k_{35} = 1$. The free parameters (i.e., $\tau_4, \tau_7, \tau_{10}, \tau_{11}$, and $\tau_{12}$) can be solved in terms of the conserved quantities. Then, a Venn diagram is used to show which among the steady state concentrations (associated with their species) are independent on specific conserved quantities. For instance, the steady state concentrations of species $X_{13}, X_{14}$, and $X_{15}$ do not depend on the conserved quantities $x_{16} + x_{17} = x_{16,0} + x_{17,0}$ (violet) and $x_{18} + x_{19} = x_{18,0} + x_{19,0}$ (yellow); that is why these species are placed inside the violet and the yellow circles. Importantly, the steady state concentrations of species $X_2, \ldots, X_8$, and $X_{20}$ do not depend on all the initial conditions and conserved quantities (the species are placed inside all the circles), but the steady state concentration of species $X_{21}$ depends on all the conserved quantities (the species is placed outside all the circles).

Previously, enormous numbers of numerical simulations were performed to investigate the bistability of the CRISPRi toggle switch model (Fig 3) even within a limited range of parameters [26]. However, in this work, the closed form of its steady state, derived with our approach, allows us to easily confirm the absence of bistablility. Specifically, the closed form allows the flexibility of investigating the steady state concentrations of various species with respect to varying initial concentrations and parameters. This elucidates the monotonicity of the steady state concentration of $S$ (single guide RNA 1) over varying the rate constants, which confirms that the model could not produce bistability in species $S$. Hence, this predicts that additional reactions had to be introduced to the network to capture the experimentally observed bistability in the system [26].

Furthermore, we were able to easily obtain the analytic steady state of the insulin model with the help of COMPILES, which allows us to quickly detect which species have absolute concentration robustness (ACR). That is, our method identifies those species whose steady state concentrations are maintained within a narrow range, regardless of the changes in the amounts of other species in the network [14].

Solving steady states numerically is common for establishing multistability, performing sensitivity analysis, conducting bifurcation analysis, and determining steady state stability. Numerical approaches, however, typically involve an enormous amount of computation and investigate a limited range of parameter values. Solving these states analytically, on the other hand, allows these procedures to be done efficiently and within a wider range of parameters.

Our work focuses on the derivation of steady states and its usefulness in analyzing biochemical systems. It would be an interesting future work to extend our framework to analyze the other long term behaviors of networks, such as stability of steady states [31–33], boundary steady states [34, 35], and oscillations [36]. In particular, for linear stability analysis of the complex-balanced equilibria of generalized networks, Boros et al. [37] and Müller and Regensburger [38] have recently proposed interesting approaches.

While we focus on the steady states of deterministic systems, the stationary distributions of stochastic systems can also be derived analytically for WR and DZ biochemical reaction networks [23, 39–41]. It would be interesting in future work to investigate whether the combination of network decomposition and translation can be used to derive stationary distributions analytically for a large class of biochemical reaction networks.

## Methods

### Chemical reaction networks

A *chemical reaction network* (CRN) can be seen as a finite collection of unique *reactions*. For instance, the following network, denoted by $\mathcal{N}$, is a CRN with four reactions:

$$R_1 : 0 \to A \qquad R_3 : A + B \to B$$
$$R_2 : 0 \to B \qquad R_4 : B \to 0$$

and fundamental units $A$ and $B$ called *species*.

In this CRN, reactions $R_1$ and $R_2$ indicate the production of species $A$ and $B$, respectively. Additionally, $R_3$ signifies that the encounter between $A$ and $B$ results in the disappearance of $A$. Finally, $R_4$ designates the consumption of $B$.

The structure of a CRN can be easily viewed as a directed graph where the edges are the reactions, and the *nodes* are non-negative linear combinations of the species $A$ and $B$ in the network. In particular, $R_3 : A + B \rightarrow B$ has nodes $A + B$ and $B$, which are called the *source* node (before the arrow) and the *product* node (after the arrow), respectively. Hence, we are in a position to say that a CRN is composed of these three sets: the sets of species, nodes (also called *complexes*), and reactions [7, 25].

We associate each reaction of a CRN with the difference between its product and source nodes called a *reaction vector*. Thus, the reaction vectors for $R_1$, $R_2$, $R_3$, and $R_4$ are given as follows:

$$(A) - (0) = A = [1, 0]^\top \quad (B) - (A + B) = -A = [-1, 0]^\top$$
$$(B) - (0) = B = [0, 1]^\top \quad (0) - (B) = -B = [0, -1]^\top$$

by identifying the two species $A$ and $B$ with the standard basis column vectors $[1, 0]^\top$ and $[0, 1]^\top$ of the Euclidean space $\mathbb{R}^2$, respectively. Then, the *stoichiometric matrix* of $\mathcal{N}$ is a matrix whose columns are the reaction vectors, and is written as

$$
\begin{array}{c}
\phantom{A} \\
A \\
B
\end{array}
\begin{array}{cccc}
R_1 & R_2 & R_3 & R_4 \\
\begin{bmatrix} 1 & 0 & -1 & 0 \\ 0 & 1 & 0 & -1 \end{bmatrix}
\end{array},
$$

whose rank is given by $s = 2$.

With the rank of this matrix, we can define a very important concept in CRN theory called the *deficiency* of a CRN [8–10], denoted by $\delta$, which can be easily calculated using the formula $\delta = n - \ell - s$, where $n$ is the number of nodes, $\ell$ is the number of connected components, and $s$ is the rank of the stoichiometric matrix. The deficiency of a CRN is a non-negative integer that can be interpreted as the measure of linear dependence among its reactions. Networks with $\delta = 0$ have the highest linear independence among the reactions. On the other hand, a high value of $\delta$ of a CRN means that linear independence is low [15].

A CRN is usually endowed with a kinetics to describe the evolution of the concentration of the species over time, which forms the *chemical reaction system*. Specifically, when the kinetics is *mass-action*, the rate function of each reaction is proportional to the the product of the concentration of the species in its source node. Hence, this rate function is a proportionality constant (*rate constant*) times the product of each concentration raised to the stoichiometric coefficient of the species that occurs in the source node of the associated reaction.

Suppose that $a$ and $b$ are the concentrations of the species $A$ and $B$, which evolve over time. Then, the rate functions are given as follows:

$$
\begin{aligned}
R_1 &: 0 \rightarrow A & k_1 a^0 b^0 &= k_1 \\
R_2 &: 0 \rightarrow B & k_2 a^0 b^0 &= k_2 \\
R_3 &: A + B \rightarrow B & k_3 a^1 b^1 &= k_3 ab \\
R_4 &: B \rightarrow 0 & k_4 a^0 b^1 &= k_4 b
\end{aligned}
$$

where $k_i$ is the rate constant of the reaction $R_i$.

Thus, the set of ODEs that describes the dynamics of the mass action system is the following:

$$
\begin{bmatrix} \dfrac{da}{dt} \\[2mm] \dfrac{db}{dt} \end{bmatrix} = k_1 \begin{bmatrix} 1 \\ 0 \end{bmatrix} + k_2 \begin{bmatrix} 0 \\ 1 \end{bmatrix} + k_3 ab \begin{bmatrix} -1 \\ 0 \end{bmatrix} + k_4 b \begin{bmatrix} 0 \\ -1 \end{bmatrix}
$$

where $\dfrac{da}{dt}$ and $\dfrac{db}{dt}$ are the time derivatives of the concentration functions of the species $A$ and $B$, respectively. Furthermore, the equation can be written as

$$
\frac{da}{dt} = k_1 - k_3 ab \text{ and } \frac{db}{dt} = k_2 - k_4 b.
$$

The long-term behavior of a system is often described by *steady states*, which could be solved by equating each time derivative to zero, i.e., $\dfrac{da}{dt} = \dfrac{db}{dt} = 0$, and solving for the concentrations of the species in terms of the rate constants. Hence, $k_1 - k_3 ab = 0$ and $k_2 - k_4 b = 0$, so the steady state is $(a, b) = \left( \dfrac{k_1 k_4}{k_3 k_2}, \dfrac{k_2}{k_4} \right)$ where $k_1, k_2, k_3, k_4 > 0$. Alternatively, the steady state can be observed from a simulation of the ODEs but for particular rate constants of the reactions and specific initial values of the concentrations of the species. Although, in general this approach is used, especially when the network is large and complicated, because it is easier to simulate rather than derive the closed form of the steady state, important properties of steady states such as existence and uniqueness are difficult to justify using numerical simulations.

## Network decomposition

A *decomposition* of a CRN is induced by a partition of its reaction set. Suppose that we partition the reaction set $\mathcal{R} = \{R_1, R_2, R_3, R_4\}$ of the CRN $\mathcal{N}$ into the following subsets:

$$
\mathcal{R}_1 = \{R_1, R_3\} \quad \text{and} \quad \mathcal{R}_2 = \{R_2, R_4\}.
$$

This partition then induces a decomposition of the network into two subnetworks, which we label $\mathcal{N}_1$ and $\mathcal{N}_2$.

In the case where the rank of stoichiometric matrix of the whole network is the sum of the ranks of the stoichiometric matrices of its subnetworks, then decomposition is called *independent* and we refer to the subnetworks as *independent subnetworks* [24, 25]. Specifically, the stoichiometric matrices of $\mathcal{N}_1$ and $\mathcal{N}_2$ are

$$
\begin{array}{c} \begin{matrix} R_1 & R_3 \end{matrix} \\ \begin{matrix} A \\ B \end{matrix} \begin{bmatrix} 1 & -1 \\ 0 & 0 \end{bmatrix} \end{array} \text{ and } \begin{array}{c} \begin{matrix} R_2 & R_4 \end{matrix} \\ \begin{matrix} A \\ B \end{matrix} \begin{bmatrix} 0 & 0 \\ 1 & -1 \end{bmatrix} \end{array}, \text{ respectively.}
$$

The ranks of these matrices are both one, i.e., $s_1 = s_2 = 1$. Since the rank of the entire network is $s = 2 = 1 + 1 = s_1 + s_2$, then the decomposition is independent. In general, to obtain the finest independent decomposition (the independent decomposition with the maximum number of subnetworks), we follow the method introduced in [42, 43] (see the Supplementary Methods in S1 Text for details).

It was shown in [24, 25] that when the underlying network decomposition of a reaction network is independent, then the set of positive steady states of the whole system is equal to the intersection of the sets of positive steady states of the subsystems as long as the same kinetics is followed by each reaction from the whole network down to its corresponding subnetwork. The theorem is given as follows (see the Supplementary Methods in S1 Text for details).

**Theorem 1** *Let $\mathcal{N}$ be a reaction network with kinetics $\mathcal{K}$ decomposed into subnetworks $\mathcal{N}_1, \mathcal{N}_2, \ldots, \mathcal{N}_\alpha$ and $\mathcal{K}_i$ be the restriction of $\mathcal{K}$ to reactions in $\mathcal{N}_i$. Then*

$$E_1 \cap E_2 \cap \cdots \cap E_\alpha \subseteq E$$

*where E is the set of positive steady states of the whole network while $E_i$ is the set of positive steady states of subnetwork $\mathcal{N}_i$. Furthermore, if the network decomposition is independent, then equality holds, i.e.,*

$$E_1 \cap E_2 \cap \cdots \cap E_\alpha = E.$$

## Generalized chemical reaction networks

In this section, we formally define the important concept of generalized chemical reaction networks (GCRNs) pioneered by Müller and Regensburger [12, 44].

Let $G = (V, E)$ be a directed graph with vertex set $V$ and edge set $E \subseteq V \times V$. Additionally, let $V_s = \{i | i \to j \in E\}$, which is the set of all the source vertices.

**Definition 1** *A generalized chemical reaction network (GCRN) is a directed graph $G = (V, E)$ together with two maps*

1. *$y : V \to \mathbb{R}^m_{\geq 0}$ that assigns to each vertex a stoichiometric complex, and*

2. *$\tilde{y} : V_s \to \mathbb{R}^m_{\geq 0}$ that assigns to each vertex a kinetic complex.*

Given a CRN with an associated graph $G$, a dynamically equivalent GCRN (i.e., the associated ODEs of the GCRN agree with the ODEs of the original CRN) is a graph $G'$ together with two maps $y$ and $\tilde{y}$, and hence, with two sets of complexes. It gives rise to two associated CRNs, the stoichiometric one $(G', y)$ and a kinetic-order one $(G', \tilde{y})$. We call the deficiencies associated to the stoichiometric CRN and kinetic-order CRN, the *effective deficiency* and *kinetic deficiency*, respectively. If there is a map between the reactions of the original CRN and the stoichiometric CRN that preserves the reaction vectors and relates the source complexes in the original CRN to the kinetic complexes in the kinetic-order CRN, a GCRN is a *network translation* of a given CRN (see [13, 20]).

We now introduce the following definition of a phantom edge and then proceed with the important result of *parametrization of positive steady states* introduced by Johnston et al. (Theorems 14 and 15 of [20]) specifically in the sense of translated networks.

**Definition 2** *For a given GCRN, we call an edge that connects identical stoichiometric complexes a* phantom edge. *Otherwise, we call it an* effective edge.

A phantom edge does not contribute to the associated ODEs because the associated stoichiometric complexes are identical. Hence, the associated dummy reaction rate constant, denoted by $\sigma$, can be arbitrary and can be considered a *free parameter*. We denote the sets of phantom edges and effective edges of the GCRN by $E^0$ and $E^*$, respectively.

**Theorem 2** *Consider a weakly reversible translated network, which is a GCRN. Let $\mathcal{F}$ be any spanning forest containing all the nodes of the kinetic-order CRN. For each edge of $\mathcal{F}$, we define the kinetic difference as the vector produced by subtracting the head kinetic complex by the tail kinetic complex. Furthermore, M is the matrix containing all the kinetic differences as rows*

*where the entries per row are arranged according to the order of the species. Let H be a generalized inverse of M (i.e., MHM = M). Finally, define B such that* im $B$ = ker $M$ *and* ker $B$ = {**0**}. *Then, if the kinetic deficiency is zero, it follows that the set of parametrized complex-balanced equilibria is given by*

$$\bar{Z} = \left\{ \kappa(k^*, \sigma)^{H^\top} \circ \tau^{B^\top} \big| \sigma \in \mathbb{R}_{>0}^{E^0}, \tau \in \mathbb{R}_{>0}^{m-\bar{s}} \right\} \neq \varnothing$$

*where* $\kappa(k^*, \sigma)^{H^\top} \circ \tau^{B^\top}$ *is the Hadamard product (the number of components of $\kappa$ is number of edges in a spanning forest, which can be effective or phantom) with the component of $\kappa$ associated with the edge $i \to i'$ as* $\kappa_{i \to i'} = \dfrac{K_{i'}}{K_i}$ *and tree constant $K_i$ as the sum (over all the spanning trees of the kinetic-order CRN towards node i) of the products of the rate constants associated with the edges of each spanning tree. In addition, if the effective deficiency is zero, then the set of positive steady states of the original network is precisely $\bar{Z}$.*

The theorem stated above covers the case when the kinetic deficiency is zero. For generalized networks with positive kinetic deficiency, additional conditions need to be checked before a steady state parametrization may be constructed (see Theorem 15 of [20]).

## Computational package, COMPILES

We developed a user-friendly, open-source, and publicly available computational package, COMPILES, that automatically decomposes a CRN into its finest independent decomposition. The package then derives the steady state of each subnetwork using the method outlined in Figs 1b and 2. Finally, COMPILES combines these subnetwork solutions to output the analytic steady state solution for the entire network in terms of the rate constants and free parameters. It also gives additional information by enumerating the conservation laws of the system.

To efficiently solve each subnetwork, a subnetwork that is already weakly reversible and of deficiency zero is no longer translated. COMPILES does its best to output the most simplified analytic solution in terms of rate constants and free parameters. If this is not feasible, the unsimplified solution (with non-free parameters) is returned.

## Supporting information

**S1 Text. Supplementary methods, Tables A-B, and Figs A-B.**
(PDF)

## Acknowledgments

The authors acknowledge Hyukpyo Hong and Yunmin Song for helpful discussions.

## Author Contributions

**Conceptualization:** Bryan S. Hernandez, Patrick Vincent N. Lubenia, Matthew D. Johnston, Jae Kyoung Kim.

**Funding acquisition:** Jae Kyoung Kim.

**Methodology:** Bryan S. Hernandez, Patrick Vincent N. Lubenia, Matthew D. Johnston, Jae Kyoung Kim.

**Project administration:** Jae Kyoung Kim.

**Software:** Patrick Vincent N. Lubenia.

**Supervision:** Jae Kyoung Kim.

**Visualization:** Bryan S. Hernandez.

**Writing – original draft:** Bryan S. Hernandez, Patrick Vincent N. Lubenia, Matthew D. Johnston, Jae Kyoung Kim.

**Writing – review & editing:** Bryan S. Hernandez, Patrick Vincent N. Lubenia, Matthew D. Johnston, Jae Kyoung Kim.

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
