## [Decision Letter · Decision Letter 0]

11 Jan 2023

Dear %TITLE% Kim,

Thank you very much for submitting your manuscript "A framework for deriving analytic long-term behavior of biochemical reaction networks" for consideration at PLOS Computational Biology. As with all papers reviewed by the journal, your manuscript was reviewed by members of the editorial board and by several independent reviewers. The reviewers appreciated the attention to an important topic. Based on the reviews, we are likely to accept this manuscript for publication, providing that you modify the manuscript according to the review recommendations.

Sincerely,

Attila Csikász-Nagy

Academic Editor

PLOS Computational Biology

Lucy Houghton

Staff

PLOS Computational Biology

Reviewer's Responses to Questions

**Comments to the Authors:**

Reviewer #1: The manuscript "A framework for deriving analytic long-term behavior of biochemical reaction networks" describes a mathematical approach for calculating the analytic steady states of a reaction network, realized as a system of (nonlinear) differential equations with polynomial right-hand side.

The manuscript is exceptionally well written, with many illuminating examples, clear explanations, and strong motivation. The mathematical approach is supported by the authors' new computational package, "COMPLILES".

The manuscript makes significant progress in addressing an important and difficult mathematical/computational problem, and I strongly recommend that it be accepted for publication.

Minor suggestion: in several places in the manuscript the general description of the importance of calculating the steady staes analytically is based on the fact that "long-term behaviors of biochemical systems are described by their steady states". While this is of course true for many reaction networks, it may not be applicable for the cases where there are other attracting sites, such as oscillations, or boundary steady states, etc. I suggest the minor change that "long-term behaviors of biochemical systems are often described by their steady states" (I inserted the word "often", to suggest that there are some exceptions).

Reviewer #2: In the manuscript “A framework for deriving analytic long-term behavior of biochemical reaction networks”, Hernandez et al. have a computational framework to analyze a molecular interaction network by breaking it into smaller independent subnetworks to derive steady states of the system. Authors have applied their computational tool to analyze toggle switch and insulin signaling models.

It is very useful method and the computational tool which can be broadly applied. Thus, this work is well suitable for PLoS Computational Biology readers.

I have only minor suggestions that may improve the manuscript.

if there are limitations when the molecular interaction network cannot be analyzed with this approach or limits for the computational package applicability, I suggest to add such information to the discussion section.

In line 101: “B+C->C”, should it be B+C->B as Figure 1 shows this reaction?

Reviewer #3: The review is uploaded as an attachment.

**Have the authors made all data and (if applicable) computational code underlying the findings in their manuscript fully available?**

Reviewer #1: Yes

Reviewer #2: Yes

Reviewer #3: Yes

PLOS authors have the option to publish the peer review history of their article (what does this mean?). If published, this will include your full peer review and any attached files.

Reviewer #1: No

Reviewer #2: **Yes: **Pavel Kraikivski

Reviewer #3: No

Figure Files:

Data Requirements:

Reproducibility:

References:

---

## [Decision Letter · Decision Letter 1]

6 Mar 2023

Dear %TITLE% Kim,

Thank you very much for submitting your manuscript "A framework for deriving analytic steady states of biochemical reaction networks" for consideration at PLOS Computational Biology. As with all papers reviewed by the journal, your manuscript was reviewed by members of the editorial board and by several independent reviewers. The reviewers appreciated the attention to an important topic. Based on the reviews, we are likely to accept this manuscript for publication, providing that you modify the manuscript according to the review recommendations.

Please correct the few minor errors the reviewer listed below.

Sincerely,

Attila Csikász-Nagy

Academic Editor

PLOS Computational Biology

Lucy Houghton

Staff

PLOS Computational Biology

Reviewer's Responses to Questions

**Comments to the Authors:**

Reviewer #3: The authors have addressed my major concerns and revised the manuscript accordingly.

I recommend the manuscript for publication after fixing some remaining minor problems.

line 184: please modify as follows:

Then, for each edge of the tree, find the ratio of the tree constants (center) and the kinetic difference (middle right), for example, the ratio of ... , and the associated kinetic difference -1A+ ...

line 188: remove transpose (it is a row vector according to your convention)

Fig 2 d: see my comments on tau_1^0, below.

line 198: the kernel of M is trivial, that is, has dimension zero, so there is no free parameter tau_1.

#free param = dim(ker(M))

The role of free monomial parameters is better explained in network N_1.

line 311: with mass-action kinetics

line 315: enormous numbers

line 333: an enormous amount

line 339: stability of steady states

line 342: Boros et al [] and Muller and Regensburger [] have recently proposed interesting approaches.

line 409: of generalized chemical reaction networks

line 415: that assigns to each vertex a stoichiometric complex

line 416: that assigns to each vertex a kinetic complex

Theorem 2: the set of parametrized complex-balanced equilibria

line 436: (the number of components of κ is the total number of effective and phantom edges)

this is not correct.

#components of kappa = number of edges in a spanning forest (which can be effective or phantom)

**Have the authors made all data and (if applicable) computational code underlying the findings in their manuscript fully available?**

Reviewer #3: Yes

PLOS authors have the option to publish the peer review history of their article (what does this mean?). If published, this will include your full peer review and any attached files.

Reviewer #3: No

Figure Files:

Data Requirements:

Reproducibility:

References:

---

## [Editor Report · Decision Letter 2]

20 Mar 2023

Dear %TITLE% Kim,

We are pleased to inform you that your manuscript 'A framework for deriving analytic steady states of biochemical reaction networks' has been provisionally accepted for publication in PLOS Computational Biology.

Best regards,

Attila Csikász-Nagy

Academic Editor

PLOS Computational Biology

Lucy Houghton

Staff

PLOS Computational Biology

---

## [Editor Report · Acceptance letter]

4 Apr 2023

PCOMPBIOL-D-22-01802R2 

A framework for deriving analytic steady states of biochemical reaction networks

Dear Dr Kim,

I am pleased to inform you that your manuscript has been formally accepted for publication in PLOS Computational Biology. Your manuscript is now with our production department and you will be notified of the publication date in due course.

With kind regards,

Anita Estes
